# Unconventional Animal Species Participation in Animal-Assisted Interventions and Methods for Measuring Their Experienced Stress

**DOI:** 10.3390/ani14202935

**Published:** 2024-10-11

**Authors:** Éva Suba-Bokodi, István Nagy, Marcell Molnár

**Affiliations:** Institute of Animal Husbandry, Hungarian University of Agriculture and Life Sciences, Kaposvár Campus, 40. Guba S. u., 7400 Kaposvár, Hungary; suba-bokodi.eva@phd.uni-mate.hu (É.S.-B); molnar.marcell@uni-mate.hu (M.M.)

**Keywords:** animal-assisted services, AAS, AAI stress, cortisol, animal welfare, guinea pigs, birds, rabbits, reptiles, donkeys, farm animals, aquarium fish, alpacas, dolphins

## Abstract

**Simple Summary:**

The demand for animal-assisted services (AAS) has increased in the past decades. The participation of supporting animals in complementary therapy for humans is a developing area. While dogs and horses are the most widely utilized species in AAS, several unconventional species have already been involved, although there is a lack of information about the effects of the stress they may experience. During the interventions, animals may experience stress that can potentially violate aspects of animal welfare. This review seeks to identify criteria for selecting unconventional animal species, such as guinea pigs, rabbits, farm animals, alpacas, donkeys, reptiles, aquarium fishes, and dolphins, for AAS, considering factors such as temperament, trainability, human–animal bond potential, stress measurement, and stress mitigation strategies. Despite the growing interest in AAS, our literature review underscores the scarcity of research exploring their effects on unconventional animal species, especially in stress measurement; therefore, more extensive studies should be conducted, for instance, measuring biochemical parameters, such as cortisol.

**Abstract:**

The participation of animals during complementary therapy for humans is a developing area. Dogs and horses are the most frequent partner species in animal-assisted services, but several unconventional species have also been involved, although there is a lack of information about the stress they experience caused by AAS. We conducted a comprehensive literature search, analyzing 135 articles with the purpose of investigating the effects of AAS on unconventional species such as guinea pigs, rabbits, farm animals, alpacas, donkeys, reptiles, aquarium fishes, and dolphins. We found that the relevant articles emphasize investigating the impact of animal-assisted interventions on humans, and they generally report positive outcomes. Limited data is available concerning the potential consequences the interventions may have on the animals. Therefore, it is our conclusion that more extensive studies should be conducted to get adequate information on stress experienced by animals during AAS, such as the measurement of biochemical parameters such as cortisol. Hence, meeting animal welfare considerations in addition to human interests could serve as a basis for the recommended methodology for therapies.

## 1. Introduction

The long-standing relationship of humans with companion animals and its impact on health is increasingly recognized [1]. The field of human–animal interaction (HAI) has undergone a significant transformation over the past five decades. The rise of animal-assisted intervention as a complementary therapy underscores the evolving nature of human–animal interactions. We are witnessing a significant increase in the number of interdisciplinary researchers and practitioners dedicated to furthering our understanding and applications of HAI [2]. While the participation of dogs, horses, and cats has traditionally dominated animal-assisted interventions, a growing trend indicates a rising demand for the inclusion of alternative, “special” animal species. Through a literature review, this research aims to gather information on:the suitability of different animal species for diverse AAS roles, considering factors such as temperament, trainability, and human–animal bond potential;the selection process for individual animals within species;methodologies for measuring stress in animals participating in AAS;potential strategies for minimizing the stress during AAS.

In this review article, we present the findings of a comprehensive literature search conducted using articles by academic publishers that have been openly accessible online in the English language for the past two decades and are available on Google Scholar. The bibliographies of the articles were also examined, and sources that directly addressed the research question were then incorporated into the study. The keywords were: the species of the animal specified in the subsection, animal-assisted service (AAS), animal-assisted special programs (AASP), animal-assisted activity (AAA), animal-assisted interventions (AAI), animal-assisted therapy (AAT), animal-assisted education (AAE), animal-assisted pedagogy (AAP), anxiety, stress. In total, 135 articles were processed with the purpose of investigating the effects of AAS on unconventional species such as guinea pigs, rabbits, farm animals, alpacas, donkeys, reptiles, aquarium fishes, and dolphins.

## 2. Different Types of Animal-Assisted Services

There are different forms of Animal-assisted services (AAS). According to the International Association of Human–Animal Interaction Organizations [2], animal-assisted interventions represent goal-oriented programs that utilize human–animal interactions to enhance well-being within healthcare, education, and social services [2]. The term AAS is often confused with the terms animal-assisted treatment (AATx) or animal-assisted support programs (AASP) [2]. The handler of the animal in all cases must be qualified for this type of work and able to train and supervise the animal during the session. The goals of the treatment are set, continuously controlled, and finally evaluated [2,3,4,5]. However, AASP is a less formal intervention in which goals are not strictly clarified. Simply the presence of the animal creates an intimate atmosphere by the action of getting in contact with the animal, which may fulfill the aims of the intervention and can be a successful healing treatment in complementary therapy [6]. Ensuring ethical practice in animal-assisted services (AAS) necessitates appropriate qualifications and experience for all participants, including human professionals, animal handlers, and the animals themselves [7].

Another popular form of AAS is animal-assisted education (AAE). Working with animals—especially dogs—increases the motivation of children in a classroom environment [8] and contributes to empathy [9,10]. Those primary school pupils who had regularly been in contact with animals in school showed higher self-confidence in their own ability to solve tasks successfully. The pupils reported that their relationships with their peers were more positive, and the interactions between them increased. Similarly, their communication skills improved, and they became more sensitive to recognizing inappropriate behavior [8].

The participation of animals in complementary therapy for humans is a developing area [11,12,13,14] as they appear in acute and hospice care, kindergartens, primary schools, veteran homes, and homes for the elderly [15]. While the benefits of AAS for humans are becoming more evident, little is known about its benefits for the animals. Ensuring animal welfare and health in all aspects of AAS is a basic requirement [16]; therefore, handlers and all participants who are involved in the interventions are required to be appropriately qualified and experienced, licensed professionals [2,7]. During the interventions, animals may experience physical and mental stress that can violate animal welfare [16]. Exposure to stress can influence many physiological processes, including those of the immune, nervous, and endocrine systems [17]. Chronic stress can induce a cascade of physiological maladaptation that can manifest in long-term adverse health effects such as depression, anxiety, allergies, and cancer [3,18]. It is the handler’s responsibility that the animals do not experience any mistreatment or harm [3].

### 2.1. Various Animal Species Participation in AAS

In the initial years of AAS, farm animals basically served to improve human health [16]. The first documented participation of animals in a therapeutic environment was recorded in the late 18th century in England. In William Tuke’s psychiatric hospital, inmates took care of rabbits and chickens belonging to the institution’s garden [19,20]. The need to own companion animals is increasing. In 2020, the most common companion animal species in the European Union were cats, with a population of approximately 75.3 million (M); dogs were the second most popular with 65.5M, followed by ornamental birds (35.6 M) and small mammals (19.4 M) [21]. According to data from statista.com, in 2022, the number of pet mammals significantly increased in the EU by more than 50%: cats, 127.1 M; dogs, 104.3 M; and small mammals, 29.3 M [22]. Nowadays, dogs are the most popular therapy animal species [23]. In the last three decades, several other species have also participated, such as guinea pigs [24,25], birds [23,26], rabbits [10,15,27,28,29], reptiles [30], donkeys [31], farm animals like dairy cows, sheep, horses, pigs, and poultry [31,32,33,34], aquarium fish [26], alpacas [35], and dolphins [36]. The participation of unconventional animals in AAS is not advised by Menna et al. [37] as these species are not closely related to humans based on the ethological level. The most frequent animal partners in AAS are dogs or horses who are at a higher level of domestication; they often seek the company of humans, while in most cases, unconventional animal species only tolerate human company [37].

#### 2.1.1. Guinea Pigs

Guinea pigs are frequently kept as companion animals [38]. However, diverse perspectives exist concerning their social interactions with humans. Johansson [39] states that humans domesticated guinea pigs and they have become social animals with more interactions. O’Haire [38] also chose guinea pigs for AAS because they are relatively easy to handle and care for, they rarely bite, and generally like to be held. On the contrary, Gut et al. [25] claim that guinea pigs rarely behave socio-positively; furthermore, it is not natural for them to be stroked or touched. Guinea pigs demonstrate a strong preference for concealed spaces. Therefore, providing a suitable hideout is essential to minimize stress and anxiety-related behaviors.

Table 1 presents a compilation of data gathered from English language, open-access research conducted within the past two decades on Google Scholar, focused on the involvement of guinea pigs in AAS and including information on the human target group, the effects of AAS on both humans and guinea pigs, and the nature of these effects (positive or negative). The keywords were: guinea pig, AAS, AAI, AAT, animal-assisted education (AAE), animal-assisted pedagogy (AAP), anxiety, and stress.

Gut et al.’s [25] study was the only one examining the effect interaction with humans has on the behavior or well-being of guinea pigs. The AAS program of Gut involved five guinea pigs, and he made 50 observations using repeated measurements. In order to ensure animal welfare for therapy guinea pigs, the possibility of a hideout is required during the sessions; otherwise, the frequency of freezing increases significantly. The research did not observe comfort behavior during therapy without the possibility of retreat.

#### 2.1.2. Birds

Owning pet birds may contribute to the well-being of human companions, potentially leading to mental and physical health improvements. Studies suggest that pet bird ownership can be associated with reduced blood pressure and lessened symptoms of depression [43].

The low level of domestication in exotic birds [43] presents challenges for their participation in AAS. To prevent fear of humans, various techniques are in practice [43,44,45]:

Pinioning or nerve blocking: this method allows for easy handling but raises ethical concerns due to its impact on the bird’s natural behavior and ability to fly [43,45].

Imprinting: while hand-rearing in the bird’s early life is effective in taming, it may lead to their own species misidentification, where the bird perceives humans as conspecifics. This may cause undesirable behaviors, such as aggressive courtship displays, posing a potential threat to humans [46]

Neglected: exotic birds are highly intelligent and social creatures [43,47,48]. Insufficient attention and interaction can lead to mental distress, manifesting in behaviors like feather plucking or self-mutilation [48,49].

Birds that may be suitable for AAS often originate from tropical regions, necessitating careful attention to their environmental needs in captivity. Maintaining appropriate temperature and humidity levels is crucial for their well-being, just as providing a balanced diet tailored to their specific species is essential for optimal health [43]. Offering ample physical and mental stimulation opportunities, such as climbing structures, foraging activities, and toys, is crucial to prevent boredom and behavioral issues [43,48].

In 1975, Mugford and M’Comisky [50] identified a potential benefit of animal companionship for mental health. Their research suggested that the presence of pet birds in elderly care homes was associated with a reduction in depression among residents. Subsequent studies supported these findings, with Holcomb et al. [51] demonstrating the positive effects of birds on depression in elderly, community-dwelling individuals. Following studies also reported a decrease in depression in lonely elderly patients residing in a rehabilitation unit who were provided with budgerigars in cages for ten days [52,53].

Table 2 presents a compilation of data gathered from English language, open-access research conducted within the past two decades on Google Scholar, focused on the involvement of pet birds in AAS and including information on the human target group, the effects of AAS on both humans and pet birds, and the nature of these effects (positive or negative). The keywords were: pet birds, birds, AAS, AAI, AAT, Animal-assisted education (AAE), animal-assisted pedagogy (AAP), anxiety, and stress.

The impact of AAS on pet birds was entirely overlooked in all of the studies examined.

#### 2.1.3. Rabbits

Loukaki et al. [28] considered rabbits as an appropriate species for participation in AAS as they are playful animals, friendly to humans, easy to socialize and transport, and their body language is clear and readable. Furthermore, children can quickly bond with them because they appear in children’s literature and songs as adorable animals, and what is more, they can have contact with them when visiting the zoo or the petting farm.

In AAS programs, rabbits can be effective as therapy animals, but to meet animal welfare requirements, they need to undergo a careful screening process [5]. Granger and Kogan [54] gave several criteria on how to select rabbits for participation in AAS programs. Mallon’s [55] proposals also give the basics of rabbit-assisted interventions (RAI). The animals need to be familiar with transport boxes and be able to cope with interaction with strangers, even if being held on a patient’s lap for 2 min, as due to rabbits’ inherent prey species nature, the act of handling itself can be a significant stressor. However, negative welfare impacts associated with handling can be minimized through the implementation of appropriate techniques. This includes the elimination of methods known to impede rabbit welfare, such as tonic immobility and scruffing [56].

Suitable rabbits should tolerate being stroked or petted by several people at the same time, as well as being touched around the area of their mouth, teeth, ears, and paws by strangers. The rabbits should also be able to be comfortable with disabled people using walking aids and not be scared of sudden sound effects, noises, or loud speech. Rabbits that meet the above criteria will be great participants in several AAS [55]. To improve the rabbits’ tolerance of being touched by humans, it is essential to accustom kits to human touch. Kits that are handled by their mother’s owner for a few minutes just after sucking milk become more extrovert and open to humans throughout their lifetime. Ethologists explain this phenomenon observed in rabbits by associating the positive experience of maternal closeness with the touch of a human hand. This sensitive period lasts for five days after birth. Therefore, training rabbits for participation in therapy starts with the breeders, who need to familiarize kits with human hands [57]. Suba-Bokodi et al. [15] worked with rabbits that were bred for participation in therapy and selected for tameness by the human approach test. The rabbits received a score according to their behavior. Cooperative, friendly, and curious Rabbits received higher scores than those that were passive, motionless, and showed signs of fear. The breeding objective is to reproduce the most extroverted individuals, creating generations.

Table 3 presents a compilation of data gathered from English language, open-access research conducted within the past two decades on Google Scholar, focused on the involvement of rabbits in AAS and including information on the human target group, the effects of AAS on both humans and rabbits, and the nature of these effects (positive or negative). The keywords: rabbit, dwarf rabbit, AAS, AAI, AAT, animal-assisted education (AAE), animal-assisted pedagogy (AAP), anxiety, and stress.

Loukaki et al. [28] suggested that rabbit-assisted services should be supervised by an expert person with considerable knowledge of biology and the natural behavior of rabbits so they can recognize the signs of fear and physiological changes in the animal. In cases where the animals show signs of unexplained aggression, hiding, changes in food consumption, repeated circling within the cage, or hiding, the session should be stopped.

The study of Molnár et al. [27] examined rabbits bred for participation in AAS, accustomed to human touch by handling them during their sensitive period immediately after birth and trained according to Mallon’s instructions. The rabbits’ behavior was monitored during the six-week-long examination period, but no signs of stress were observed. As the animals moved to the classroom, they became familiar with it. During the days when the teacher was present in the classroom, the rabbits could freely choose whether to stay inside or outside of the cage. It was observed that rabbits preferred to stay outside, move around freely, and only use their cage for food consumption, resting, or defecation and urination.

According to Součková et al. [58], ordinary dwarf rabbits bred and raised in normal pet rabbit circumstances participated in the services. Součková et al. [58] found that the rabbits showed negative emotional responses during AAS sessions, which escalated when they were placed into the lap of children. These rabbits showed a higher incidence of the freeze response. To reduce the stress level of rabbits during AAS, Součková et al. [58] recommend to use a special therapy table with the possibility of hiding.

#### 2.1.4. Farm Animals in AAS

Green care programs, so-called “care farming” or “social farming”, is an innovative sector in which healthcare and agriculture work together in order to offer supported workplaces, often with residential care to patients with different disabilities such as troubled youths, intellectual disability, mental disorders and people with dementia. In green care programs, patients are involved in agricultural production and farm-related activities [59]. Berget et al. [32] claim that AASP in “care farming” has a positive impact on the patients’ self-efficacy and increases the ability to cope with people’s long-lasting psychiatric symptoms. Mallon [55] found that children in residential treatment centers found farm animals therapeutic. Children often visited the animals, talked to them without fear, and felt better in their company. They also learned about the nature of the animals and about their care, feeding, and living.

Table 4 presents a compilation of data gathered from English language, open-access research conducted within the past two decades on Google Scholar, focused on the involvement of farm animals in AAS and including information on the human target group, the effects of AAS on both humans and farm animals, and the nature of these effects (positive or negative). The keywords were: farm animals/stock animals in AAS, AAI, AAT, AAE, AAP, anxiety, and stress.

The impact of AAS on farm animals was entirely overlooked in all of the studies examined.

#### 2.1.5. Alpaca

Alpacas’ temperament and the ability to be trained make them suitable for participation in AAS, and their appearance is considered pleasant and delightful. Children and adults also gladly get into physical contact with them, although they are not appropriate to ride on. In order for alpacas to participate in therapy, it is necessary to meet the basic requirements, such as being able to cope with strangers and tolerate when people touch them, even on their hind limbs. The animals should be familiar with walking on a leash and being transported in livestock trailers [61]. A Pilot Study was conducted by Watkinson et al. [62] at James Madison University, Virginia, USA, 2022, with the result that alpaca-assisted activities reduced the anxiety levels of college students. The influence of alpaca-assisted activities on individuals with a dual diagnosis of Down syndrome and autism spectrum disorder has been shown to be positive and beneficial, improving the involved people’s social and emotional behavior [63]. Participation in alpaca-assisted activities was also reported positively on New England Farms, as lower anxiety and depression were determined afterward. This suggests that alpacas could be a valuable addition to promoting emotional well-being [64].

Table 5 presents a compilation of data gathered from English language, open-access research conducted within the past two decades on Google Scholar, focused on the involvement of alpacas in AAS and including information on the human target group, the effects of AAS on both humans and alpacas, and the nature of these effects (positive or negative). The keywords were: alpaca in AAS, AAI, AAT, AAE, AAP, anxiety, and stress.

The impact of AAS on alpacas was entirely overlooked in all of the studies examined.

#### 2.1.6. Donkey

Keeping donkeys as companion animals is becoming more fashionable, in addition to their role as therapy animals [65]. The donkey-assisted service is called onotherapy. Their participation in several forms of AAS is inevitably a great support for children, for people with mental and emotional disorders [66], and for elderly people. Donkeys can be ridden [66]; therefore, those patients who are afraid of horses because of their size may find pleasure in donkeys as they are smaller and slower than horses [67]. Their auditory stimuli are less sensitive than horses’ [68], and people also find pleasure in touching their long and soft coats [67]. The improvement in patients’ communication had been detected [68]. Donkeys are considered gentle-natured animals [31], although they are frequently called “stubborn” [68]. Donkeys are quite trainable animals, and with professional handlers, they are willing to cooperate [68]. They are able to motivate children, which contributes to the psycho-affective and psycho-cognitive development process [69].

Table 6 presents a compilation of data gathered from English language, open-access research conducted within the past two decades on Google Scholar, focused on the involvement of donkeys in AAS and including information on the human target group, the effects of AAS on both humans and donkeys, and the nature of these effects (positive or negative). The keywords were: donkey in AAS, AAI, AAT, AAE, AAP, anxiety, and stress.

Although donkeys are generally considered to be calm animals, individual habitual differences appear, and it is incorrect to generalize this statement for all donkeys [65]. To ensure the suitability of a donkey for onotherapy, Gonzalez-De Cara et al. [68] developed a behavior test.

Clancy et al. [65] ensured animal welfare by giving donkeys the opportunity to decide whether they want to participate in the therapy or not. This develops the empathetic skills of patients who learn not to force their will on any creature. The authors also emphasized that past experience, training, and current environment may significantly impact the animals’ instantaneous calm behavior response.

#### 2.1.7. Reptiles

Keeping reptiles as pets is quite common, as millions of them can be found in households. The benefits of ordinary pet animals on the well-being of humans are well-researched, but these results may not apply to reptiles [72]. An important factor in determining human attitudes toward reptiles is what kind they are. Snakes are considered the most frightening and disliked species, while turtles are particularly popular [73]. People with narcissistic and borderline personality disorders can also establish links with exotic pets [74]. A significant knowledge gap exists regarding the potential of reptiles as service animals. Currently, no established organizations evaluate or register reptiles for such a designation. Furthermore, the existing literature offers minimal research on two crucial aspects: (1) our ability to identify stress indicators in reptilian species accurately, and (2) the capacity of these animals to provide informed consent for the specific demands of service animal roles.

In Sarman’s [75] research, almost a hundred 5 and 6-year-old children participated. They underwent blood samples being taken and catheterization at an outpatient clinic. The aim of the study was to reduce their pain during the medical examination by introducing and observing turtles and goldfish. The animals prevented children from concentrating on the medical intervention and, therefore, reduced their anxiety and fear, and their attitudes toward pain improved.

In Murry’s [30] study, 40 children participated who all experienced the death of a parent within a year. Children were offered AAS as a complementary therapy, and the animal of choice of the participants were reptiles. The children’s aggression or delinquent behavior decreased while their social behavior, attention, and thoughts improved.

Female turtles typically do not exhibit parental care behaviors after laying their eggs. This biological phenomenon could be employed therapeutically to facilitate discussions with foster children regarding their own parental experiences. The self-sufficiency observed in young turtles after hatching could offer a metaphor for foster children. Their sense of hope for their ability to develop independence and resilience despite experiencing parental separation is supported by the example of the turtles [76].

Table 7 presents a compilation of data gathered from English language, open-access research conducted within the past two decades on Google Scholar, focused on the involvement of reptiles in AAS and including information on the human target group, the effects of AAS on both humans and reptiles, and the nature of these effects (positive or negative). The keywords were: reptiles in AAS, AAI, AAT, AAE, AAP, anxiety, and stress.

The impact of AAS on reptiles was entirely overlooked in all of the studies examined.

#### 2.1.8. Aquarium Fish

While fish are a common pet animal, their welfare often receives less attention compared to mammals such as dogs and cats [77]. Scientific understanding of fish cognition is evolving, with evidence pointing towards complex mental processes influencing their behavioral responses [78]. Due to the vast diversity in the origins of fish, encompassing freshwater and marine environments, their specific needs for water quality, diet, temperature, and social interactions (predator/herbivore) necessitate a meticulous approach to maintaining a healthy social aquarium [77,79]. Studies have identified relaxation and stress reduction as potential benefits of keeping fish as companion animals. These effects appear primarily linked to observing fish movements [80], and the sound of running water is a contributing factor in increasing calmness [80]. As the animals are not in physical contact with humans, there is minimal chance of zoonosis [80]. Fine [16] writes in the Encyclopedia of Psychotherapy that during dental surgery, patient’s anxiety and discomfort could successfully be reduced by watching an aquarium. Watching fish has a calming and relaxing effect that increases the patients’ calmness.

Table 8 presents a compilation of data gathered from English language, open-access research conducted within the past two decades on Google Scholar, focused on the involvement of aquarium fish in AAS and including information on the human target group, the effects of AAS on both humans and aquarium fish, and the nature of these effects (positive or negative). The keywords were: fish, aquarium, ornamental fish in AAS, AAI, AAT, AAE, AAP, anxiety, and stress.

The impact of AAS on aquarium fish was entirely overlooked in all of the studies examined.

#### 2.1.9. Dolphins in AAS

The participation of dolphins in animal-assisted services can provide a dual therapy, namely, water and animal therapy. Swimming with them is especially effective for children who suffer from attention deficit hyperactivity disorder, physical disabilities, and autism [85]. Dolphin-assisted complementary interventions may cause controversy in comparison to other forms of AAS [61]. People’s attitude to dolphins is positive due to the media and adverts [85], but their participation in therapy must be reconsidered due to their documented aggressive behavior towards swimmers [61]. Dolphins are non-domesticated mammal species and the most exotic animals involved in AAS. Marino and Lilienfeld [86,87] emphasize that there is no compelling evidence for legitimizing dolphin therapy due to the inaccuracy of the analyzed studies. All outcomes might be caused by the fleeting improvements in mood. Kapuska [61] does not suggest dolphins’ participation in AAS due to the reasons outlined above. These animals are in captivity and may not have a choice in interacting with humans.

Table 9 presents a compilation of data gathered from English language, open-access research conducted within the past two decades on Google Scholar, focused on the involvement of dolphins in AAS and including information on the human target group, the effects of AAS on both humans and dolphins, and the nature of these effects (positive or negative). The keywords were: dolphins in AAS, AAI, AAT, anxiety, and stress.

The impact of AAS on dolphins was entirely overlooked in all of the studies examined.

## 3. Stress Caused by Animal-Assisted Services

The body’s response to environmental stress activates the adrenal axes: the sympathetic–adrenal–medullary (SAM) and the hypothalamic–pituitary–adrenal cortex (HPA) systems. When mammals feel threatened, their fear intensifies and special hormones called catecholamines and glucocorticoids are released. These hormones are responsible for ensuring immediate access to energy available in the body. By monitoring levels of catecholamines and glucocorticoids, the short-term stress response of animals becomes measurable [92,93,94,95].

### 3.1. Factors That Contribute to Animal Stress

Taking animals out of their familiar surroundings and transporting them is a definite source of stress in itself [94,95]. There are only ordinary recommendations provided in relation to transporting pet animals [96]. There are many studies that deal with stress experienced by dogs during transportation. Besides behavior studies, descriptive studies can be based on the observation of the animals [97,98,99], including the so-called stress hormone examination of cortisol, are also available [100,101,102,103]. Suba-Bokodi et al. [5] examined stress caused by repeated transportation on rabbits by hormone cortisol determination. They stated that transportation had a negative impact on the animals’ stress levels and, furthermore, two weeks was not long enough to condition rabbits to the process.

AAS animals are often required in schools and hospitals where they encounter a new environment and multiple new people. In educational institutions, children’s abilities and developmental ages fall within a wide range, and their behavior may suddenly change; they can be loud and guided by their emotions. On the other hand, medical centers are often crowded and smell of intensive antiseptics and cleaning agents. When healthcare workers move quickly, the circumstances may suddenly change from calm and peaceful to urgent and intense [104].

### 3.2. Techniques for Analyzing Hormonal Indicators of Stress

Determining stress hormone concentration in blood is an obvious method, but the process of taking samples is a severe stressor itself, as animals must be captured and restrained, which can interfere with the assessment of adrenocortical responses. This especially applies to non-domestic animals [93,105,106,107]. To reduce the stress caused by taking blood samples, glucocorticoid levels can be determined from feces, urine, saliva, or hair in mammals. In fish, corticosteroids can be measured in excreta and water [93,105,106].

Glucocorticoid metabolites (GCM) accumulate in feces several hours after exposure to stress, depending on the species. This reflects the extent of stress experienced without inducing acute hormonal responses associated with collecting samples. The limitation of the method is that bacterial metabolism and enzymatic action can cause further degradation in the feces cortisol/ corticosterone concentration; therefore, the sampling and storing conditions are critical [105,106], and in the case of the animals kept in groups, samples cannot be assigned to individuals [108]. This method had already been used for determining stress in European rabbits [107] and in dwarf rabbits [5].

The collection of saliva samples presents minimal difficulty due to the nature of domesticated species like canines and equines [109,110,111,112]. The collection of saliva often necessitates handling and restraining, yet it is still less stressful than taking blood samples as the procedure takes less time [105]. Cortisol content is relatively stable in salvia at room temperature for several hours or even days. The bacteria present may multiply in the sample, leading to degradation of the stress hormone. For optimal preservation of cortisol, saliva freezing is recommended immediately after collecting samples [105]. The collection of saliva via direct sputum sampling is limited to specific species. Flavored absorbent materials such as cotton ropes/pads are used most often to stimulate licking or chewing [113].

While urinary cortisol analysis is possible, sampling difficulties limit its practicality in many instances [108]. Taking urine from the environment is difficult to implement [92], while taking it directly from the bladder causes stress similar to taking blood samples as it also requires the use of a needle [114]. Plasma cortisol displayed a rapid rise within 2 h, staying elevated ≥8 h. Urinary cortisol exhibited a similar pattern with a delayed peak (2–6 h) [115].

Hair cortisol concentration (HCC) measurement provides a non-invasive approach to chronic stress assessment in domestic and wild animals and reflects cortisol exposure over weeks. Systemic cortisol can only be incorporated into the growing hair shaft from blood vessels via passive diffusion during the anagen phase and occurs within the hair follicle, situated several millimeters beneath the skin surface. HCC analysis has a built-in time lag between cortisol incorporation and hair emergence due to variable growth rates across species and body regions and must be considered when using HCC as an indicator of stress [108].

### 3.3. Techniques for Analyzing Non-Hormonal Indicators of Stress

Acute stress responses are a vital component of survival for wild-living animals. However, the inherent difficulties associated with field-based assessments of underlying physiological processes restrict our comprehension of how variations within the acute stress response influence fitness in free-ranging animals [116]. While thermal imaging offers a non-invasive approach to measuring stress-induced changes in body surface temperature, its application in field studies is limited by the significant influence of environmental conditions on the thermal response. Several studies have employed eye temperature measurements to assess physiological responses in various species, including cows [117], buffalo bulls [118], mice [119], horses [120,121,122], cats [123], sheep [124].

Assessment of heart rate variability serves as a non-invasive tool for investigating autonomic nervous system function, with a specific focus on the balance between sympathetic and parasympathetic (vagal) activity [125]. The acute stress response triggers a cascade of physiological changes. Epinephrine binding to β-adrenergic receptors rapidly increases heart rate within seconds, followed by a gradual return to baseline levels within 5–10 min, depending on the stressor intensity. The parasympathetic nervous system, via its influence on beat-to-beat variability, modulates heart rate. During stress, heart rate not only accelerates but also exhibits reduced variability (decreased HRV) as the fight-or-flight response dominates. While both heart rate and HRV demonstrate rapid changes during acute stress, they are also susceptible to the effects of chronic stress, such as captivity and repeated stressors. This sensitivity allows these measures to be employed to assess the impact of longer-term stimuli (e.g., seasonal variations) lasting for months [126]. According to the evidence of studies published after the turn of the millennium, the examination of heart rhythm and heart rhythm variance is an accepted way of detecting stress in vertebrates [127].

Traditionally employed stress assessment techniques, such as cortisol level quantification and heart rate/variability analysis, necessitate specialized equipment and often lack immediate results; therefore, applying alternative methodologies is justified in several species. The spontaneous blink rate (SBR) represents a promising non-invasive approach, having been previously utilized in human stress response studies. Research suggests a positive correlation between SBR and established stress markers like heart rate variability and salivary cortisol levels in animals, potentially making it a rapid stress indicator. SBR has been validated as a reliable measure in horses, but the initial “startle” response warrants careful consideration to avoid misinterpretations [128]. Partials blinks have also been documented in both dogs and cats [129,130].

## 4. Conclusions

According to the comprehensive literature analysis conducted on 135 articles investigating AAS effects on unconventional species such as guinea pigs, birds, rabbits, farm animals, alpacas, donkeys, reptiles, aquarium fishes, and dolphins, we found that publications primarily focus on the AAS impact on humans, neglecting the potential consequences for animal behavior and their well-being. The articles analyzed provided valuable insights into the impact on animals’ well-being and their stress response during AAS related to three species: donkeys, rabbits, and guinea pigs. While research typically investigates either humans or animals, rabbits are a unique species that allows us to explore the multifaceted effects of animal-assisted services on both humans and animals. This approach provides valuable insights into the intricate relationship between AAS and well-being across different species. We strongly recommend carrying out further research that expands its scope beyond the human-centric focus of AAS to encompass the experiences of the different animal species that participate. The most frequent species that participate in AAS, such as dogs, horses, and cats, are at a higher level of domestication, thus, they already seek the company of humans. Unconventional animal species, in most cases, only tolerate human company; therefore, much more careful, deep, and intensive preparation of the animal is needed to allow for their successful participation in AAS. Given the availability of technologies to measure biochemical stress indicators like cortisol, a compelling opportunity exists to investigate the potential stress response in animals participating in AAS. Moreover, the successful application of these methods with stock animals suggests their potential suitability for unconventional species. The AAS field requires a paradigm shift towards a more rigorous, evidence-based approach. This necessitates robust research methodologies that prioritize animal welfare. We have an ethical responsibility to respect the life and well-being of all organisms.

## Figures and Tables

**Table 1 animals-14-02935-t001:** Participation of guinea pigs’ attendance in AAS.

Target Group	Effect of AAS to Human	Data about the Animals: Selection Methodology, Stress Measurement	Reference
psychiatric inpatients	positive	not available	Mar, 2000 [40]
correctional institution	positive	not available	Matuszek, 2010 [41]
children with autism	positive	not available	Talarovičová, 2010 [24]
AAE with children aged 5–12	positive	not available	O’Haire, 2013 [38]
children with autism	positive	not available	O’Haire, 2014 [42]
rehabilitation clinic	no information	yes *	Gut, 2018 [25]
children with autism	positive	not available	Johansson, 2020 [39]

* above detailed.

**Table 2 animals-14-02935-t002:** Pet birds’ attendance in AAS.

Target Group	Effect of AAS to Human	Data about the Animals: Selection Methodology, Stress Measurement	Reference
elderly people	positive	not available	Mugford and M’Comisky [50]
elderly people	positive	not available	Holcomb et al. [51]
elderly people	positive	not available	Jessen et al. [52]
elderly people	positive	not available	Falk et al. [53]

**Table 3 animals-14-02935-t003:** Participation of rabbits in AAS.

Target Group	Effect of AAS to Human	Data about the Animals: Selection Methodology, Stress Measurement	Reference
complementary therapy for children	positive	yes *	Loukaki et al., 2011 [28]
AAE with children aged 6–8	positive	yes *	Molnár et al., 2020 [27]
visually impaired adults	positive	yes	Iváncsik and Molnár, 2021 [29]
children	no information	yes *	Suba-Bokodi et al., 2022 [15]
children	no information	yes *	Součková et al., 2023 [58]

* above detailed.

**Table 4 animals-14-02935-t004:** Farm animals’ attendance in AAS.

Target Group	Effect of AAS to HUMAN	Data about the Animals: Selection Methodology, Stress Measurement	Reference
children in residential treatment center	positive	not available	Mallon et al., 1994 [55]
adults with mental disorders	positive	not available	Berget et al., 2008 [32]
multiply-disabled adults	partly positive	not available	Scholl et al., 2008 [60]
psychiatric disorders	positive	not available	Berget et al., 2011 [33]
adults with clinical depression	positive	not available	Pedersen et al., 2011 [34]
adults with mental disorder	positive	not available	Hassink et al., 2017 [59]

**Table 5 animals-14-02935-t005:** Participation of alpacas in AAS.

Target Group	Effect of AAS to Human	Data about the Animals: Selection Methodology, Stress Measurement	Reference
college students	positive	not available	Watkinson et al., 2022 [62]
people with Down syndrome and autism spectrum disorder	positive	not available	Stone and Denlinger, 2022 [63]
individuals experiencing depression an anxiety	positive	not available	Masse, 2024 [64]

**Table 6 animals-14-02935-t006:** Donkeys’ attendance in AAS.

Target Group	Effect of AAS to Human	Data about the Animals: Selection Methodology, Stress Measurement	Reference
children in outpatient rehabilitation units	positive	not available	De Rose, 2011 [69]
no information	no information	yes *	Gonzalez-De Cara et al., 2017 [68]
autism spectrum disorder and intellectual disability	positive	not available	Kwon et al., 2019 [70]
adults with intellectual disability	positive	not available	Colombo et al., 2020 [71]
no information	no information	yes *	Clancy et al., 2022 [65]
children with dyslexia aged 7–12	positive	not available	Corallo et al., 2023 [66]

* above detailed.

**Table 7 animals-14-02935-t007:** Participation of reptiles in AAS.

Target Group	Effect of AAS to Human	Data about the Animals: Selection Methodology, Stress Measurement	Reference
children	positive	not available	Sarman, 2024 [75]
children having lost a parent	positive	not available	Murry and Todd, 2012 [30]
foster children	positive	not available	Hellmann, 2013 [76]

**Table 8 animals-14-02935-t008:** Presence of aquarium fish in AAS.

Target Group	Effect of AAS to Human	Data about the Animals: Selection Methodology, Stress Measurement	Reference
dentist clients	neutral	not available	Lundberg et al., 2021 [81]
adults stress test	positive	not available	Spittell et al., 2019 [82]
undergraduate students	partly positive	not available	Gee et al., 2019 [83]
elderly with dementia	positive	not available	Edwards et al., 2014 [84]

**Table 9 animals-14-02935-t009:** Participation of dolphins in AAS.

Target Group	Effect of AAS to Human	Data about the Animals: Selection Methodology, Stress Measurement	Reference
children with severe disability	positive	not available	Nathanson, 1998 [88]
children with disability	positive	not available	Humphries, 2003 [89]
children with special needs	positive	not available	Dilts et al., 2011 [90]
children with severe disability	positive	not available	Breitenbach et al., 2015 [91]

## Data Availability

Data sharing is not applicable (only appropriate if no new data is generated or the article describes entirely theoretical research). No new data were created or analyzed in this study.

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
