# Peer review of "Unconventional Animal Species Participation in Animal-Assisted Interventions and Methods for Measuring Their Experienced Stress"

_animals, 2024, doi:10.3390/ani14202935_

Round 1
Reviewer 1 Report
Comments and Suggestions for Authors
This article addresses an important topic in the field of AAS - the effects of inerventions on the well-being of the animals. The paper is well-written and should be published after addressing the following issues:
Line 41 - the authors state “Through a literature review, this research aims to gather information on:
-the suitability of different animal species for diverse AAS roles, considering factors such as temperament, trainability, and human-animal bond potential
-the selection process for individual animals within species
-methodologies for measuring stress in animals participating in AAS
-potential strategies for minimizing the stress during AAS.”
However, the abstract and simple summary address just one of these aims – evaluating the effects of interventions on unconventional species. Then on line 54, it is stated “135 articles were processed with the purpose of investigating the effects of AAS on unconventional species such as guinea pigs, rabbits, farm animals, alpacas, donkeys, reptiles, aquarium fishes and dolphins.” Please revise for clarification.
Line 93 – this is a lone statement in its own paragraph, and it reads as opinionated. Please integrate into the previous paragraph.
Line 99 – “rabbits and poultry” using the animal name and the food name together is inconsistent. Especially when discussing animal well-being, you should use the name of the animal, not the food product that comes from the animal. Should be “rabbits and [chickens].”
Line 108 – the is a long list of specific animals that includes “farm animals”, which is too vague. Please specify using the name of the animals. Not everyone will have the same definition for the term.
Tables 1 – in the text it is described as “Table 1 summarizes the articles that investigate the effects of AAS on human and/or animal participants.” But the table only has a column for the effect on the human. It should have a column titled effect on the animal. The column titled “Information about animal resources methodology…” is confusing and should be relabeled or described more in the text. THIS IS TRUE FOR ALL TABLES
Line 33 – typing error…pigs. into The….
Line 151 – Neglection should be Neglect.
For the tables where the finding was that none of the studies addressed the effect of AAS on the animals, that should be stated at the end of the paragraph describing the table.
Author Response
This article addresses an important topic in the field of AAS - the effects of interventions on the well-being of the animals.
Thank you very much for your kind efforts to make our review better. All your request had been met and marked with GREEN.
The paper is well-written and should be published after addressing the following issues:
Line 41 - the authors state “Through a literature review, this research aims to gather information on:
-the suitability of different animal species for diverse AAS roles, considering factors such as temperament, trainability, and human-animal bond potential
-the selection process for individual animals within species
-methodologies for measuring stress in animals participating in AAS
-potential strategies for minimizing the stress during AAS.”
However, the abstract and simple summary address just one of these aims – evaluating the effects of interventions on unconventional species. Then on line 54, it is stated “135 articles were processed with the purpose of investigating the effects of AAS on unconventional species such as guinea pigs, rabbits, farm animals, alpacas, donkeys, reptiles, aquarium fishes and dolphins.” Please revise for clarification.
The following clarification was made in the simple summary (Line: 14-20)
This review seeks to identify criteria of selecting unconventional animal species – such as guinea pigs, rabbits, farm animals, alpacas, donkeys, reptiles, aquarium fishes and dolphins – for AAS, considering factors such as temperament, trainability, human-animal bond potential, stress measurement, and stress mitigation strategies. Despite the growing interest in AAS, our review of the literature underscores the scarcity of research exploring their effects on unconventional animal species, especially in terms of stress measurement therefore more extensive studies should be conducted, for instance the measuring biochemical parameters, such as cortisol
Line 93 – this is a lone statement in its own paragraph, and it reads as opinionated. Please integrate into the previous paragraph.
DONE! Line 96
Line 99 – “rabbits and poultry” using the animal name and the food name together is inconsistent. Especially when discussing animal well-being, you should use the name of the animal, not the food product that comes from the animal. Should be “rabbits and [chickens].”
DONE! Line 102
Note: The term of “poultry” was used because the referred article is using this form.
Line 108 – the is a long list of specific animals that includes “farm animals”, which is too vague. Please specify using the name of the animals. Not everyone will have the same definition for the term.
According to the referred articles, the “farm animal” species had been defined.
Line 111
...farm animals like dairy cows, sheep, horses, pigs and poultry...
Tables 1 – in the text it is described as “Table 1 summarizes the articles that investigate the effects of AAS on human and/or animal participants.” But the table only has a column for the effect on the human. It should have a column titled effect on the animal. The column titled “Information about animal resources methodology…” is confusing and should be relabeled or described more in the text. THIS IS TRUE FOR ALL TABLES
Rewritten in all tables:
Table 1 presents a compilation of data gathered from English language; open-access research conducted within the past two decades on Google Scholar, focuses on the involvement of guinea pigs in AAS and includes information on the human target group, the effects of AAS on both humans and guinea pigs, and the nature of these effects (positive or negative).
LINE: 127, 170, 211, 251, 276, 296, 341, 363, 386,
The column titled “Information about animal resources methodology…” is relabeled in all tables to: “Data about the animals: selection methodology, stress measurement”
Line 33 – typing error…pigs. into The….
DONE! Line: 136 (deleted “into”)
Line 151 – Neglection should be Neglect.
DONE! Line: 154
For the tables where the finding was that none of the studies addressed the effect of AAS on the animals, that should be stated at the end of the paragraph describing the table.
The following sentence is added to table 2, 4, 5, 7, 8, 9
The impact of AAS on animals was entirely overlooked in all of the studies examined.
LINES: 179, 260, 284, 349, 372, 394
Reviewer 2 Report
Comments and Suggestions for Authors
Overall, this topic is much needed. I am a certified animal assisted counselor and conducted a study with my therapy rabbit and worked with him in therapy and was constantly advocating for him since he was a smaller animal (this looked like limiting group sizes, setting physical boundaries, having his toys/hideout, educating clients on communication and body language of a rabbit, creating a protocol for warning signs/safety concerns/what to do if animal became stressed/if a client crossed a boundary of the therapy animal, etc.). This was a privilege to be aware of how to advocate for animal welfare because I was able to attend a very intensive training specially for animal assisted counseling and my animal research team at the university was very welfare focused. But a bunny is very different than the usual therapy animals (usually dogs and cats here in the U.S.) and there was little research on working with a rabbit in a therapy setting. The term Animal Assisted Counseling (AAC) is beginning to be used in America to differentiate therapists and counselors from Animal Assisted Therapy (utilized by physical therapists, occupational therapist, etc.) so it may be worth also searching this term in future.
Lines 52-53: For someone on the outside of AAI/AAC, they may not know what these acronyms are. AAI terms are also not universal so some may not know what you may mean when using various acronyms. I would spell out the word and place the acronym in parenthesis when introducing the word for the first time.
Line 60: Have you all thought of also including literature from animal assisted intervention international? (AAI-INT)? You can also reference the American Counseling Association’s Animal Assisted Therapy in Counseling Competencies when referring to/researching AAC in U.S. specifically as well.
Line 111: I would maybe change the word “used.” In animal assisted counseling (AAC), we are taught that “working with” an animal in counseling is smoother verbiage. Because as handler, we are not “using” the animal (it’s assumed that “using” is ignoring some welfare pieces not taking into consideration the animal’s perspective) and “working with” “incorporating” etc. is a collaborative relationship and team between the licensed AAC professional, the therapy animal, and of course, the client when there’s a client present.
If you all do decide to change this word, I suggest changing it throughout.
Line 189- I would even challenge “tolerate” as you want animals to enjoy what they are doing and not just tolerate it. But I understand that you also want to make sure that if a client touches an animal’s back, they will not bite. For example, clients can pet my therapy rabbit but only on his forehead where he likes to pet and that is it. There is a boundary that is set for clients. He will tolerate being pet other places and will not act aggressively but does not enjoy it and I know this by his body language. Clients are also not allowed to pick him up even though he has no issue being picked up. This is due to safety concerns as rabbits back and legs are very easily broken if a child or client were to drop them.
Line 194: Have you all thought of options outside of kits/breeding/not having the option to get a baby/young bunny? In America, there is an issue with overbreeding rabbits. I communicated with a local adoption agency for months before choosing the appropriate rabbit for Animal Assisted Counseling. I worked with the agency to adopt a rabbit that enjoyed children (they took him to visit schools), enjoyed being pet, was okay with being transported, etc.
Section 2.1.7 – My program certified a tortoise therapy team here in the U.S. in 2021!
Author Response
Overall, this topic is much needed. I am a certified animal assisted counselor and conducted a study with my therapy rabbit and worked with him in therapy and was constantly advocating for him since he was a smaller animal (this looked like limiting group sizes, setting physical boundaries, having his toys/hideout, educating clients on communication and body language of a rabbit, creating a protocol for warning signs/safety concerns/what to do if animal became stressed/if a client crossed a boundary of the therapy animal, etc.). This was a privilege to be aware of how to advocate for animal welfare because I was able to attend a very intensive training specially for animal assisted counseling and my animal research team at the university was very welfare focused. But a bunny is very different than the usual therapy animals (usually dogs and cats here in the U.S.) and there was little research on working with a rabbit in a therapy setting. The term Animal Assisted Counseling (AAC) is beginning to be used in America to differentiate therapists and counselors from Animal Assisted Therapy (utilized by physical therapists, occupational therapist, etc.) so it may be worth also searching this term in future.
Authors are grateful that based on your suggestions our review could be further improved. Ensuring Animal Welfare during AAS is just mandatory and not an option.
All your requests had been met and marked with BLUE.
Lines 52-53: For someone on the outside of AAI/AAC, they may not know what these acronyms are. AAI terms are also not universal so some may not know what you may mean when using various acronyms. I would spell out the word and place the acronym in parenthesis when introducing the word for the first time.
All the acronyms are introducing for the better understanding.
Animal Assisted Service (AAS), Animal Assisted Special Programs (AASP), Animal Assisted Activity (AAA), Animal Assisted Interventions (AAI), Animal Assisted Therapy (AAT), Animal-Assisted Education (AAE), Animal-Assisted Pedagogy (AAP),
Line 60: Have you all thought of also including literature from animal assisted intervention international? (AAI-INT)? You can also reference the American Counseling Association’s Animal Assisted Therapy in Counseling Competencies when referring to/researching AAC in U.S. specifically as well.
Thank you! This will be incorporated into the literature review for the PhD thesis of the first author.
Line 111: I would maybe change the word “used.” In animal assisted counseling (AAC), we are taught that “working with” an animal in counseling is smoother verbiage. Because as handler, we are not “using” the animal (it’s assumed that “using” is ignoring some welfare pieces not taking into consideration the animal’s perspective) and “working with” “incorporating” etc. is a collaborative relationship and team between the licensed AAC professional, the therapy animal, and of course, the client when there’s a client present.
If you all do decide to change this word, I suggest changing it throughout.
Yes, thank you. Generally we do not say “use” an animal for AAS.
The sentence is rewritten:
LINE: 22 Dogs and horses are the most frequent partner species in…
LINE: 115
The most frequent animal partners in AAS are dogs or horses who are at a higher level of domestication
LINE: 148
The low level of domestication in exotic birds [45] presents challenges for their participation in AAS
LINE: 208
Suba-Bokodi at al. [15] work with rabbits that are bred for participation in therapy and selected for tameness
LINE: 507
The most frequent species that participate in AAS such as dogs, horses…
Line 189- I would even challenge “tolerate” as you want animals to enjoy what they are doing and not just tolerate it. But I understand that you also want to make sure that if a client touches an animal’s back, they will not bite. For example, clients can pet my therapy rabbit but only on his forehead where he likes to pet and that is it. There is a boundary that is set for clients. He will tolerate being pet other places and will not act aggressively but does not enjoy it and I know this by his body language. Clients are also not allowed to pick him up even though he has no issue being picked up. This is due to safety concerns as rabbits back and legs are very easily broken if a child or client were to drop them.
The term of “tolerate” was taken from the referred article.
Absolutely! The goal is to involve animals, regardless of their species, in animal-assisted activities as long as they can enjoy it.
In my opinion, it's not the species of the animal that determines its suitability but rather the individual's personality. For example, let's take dogs as the most popular type of therapy/assistance animal. It's obvious that a breed selected for guard dog activities won't be suitable to handle the demands of animal-assisted activities.
The careful selection of animals for AAI work is essential to ensure their well-being.
Personally I have worked together with my rabbits in kindergarden, primary school, secondary school, children with mild intellectual disability and also in elderly homes. To educate human about the specie, about the rules that made for the secure of both the animals and the people, the species-specific characteristics, behavior etc. are the first steps of ensuring animal welfare just as to teach people NOT TO FORCE the animals to their willing .
Line 194: Have you all thought of options outside of kits/breeding/not having the option to get a baby/young bunny? In America, there is an issue with overbreeding rabbits. I communicated with a local adoption agency for months before choosing the appropriate rabbit for Animal Assisted Counseling. I worked with the agency to adopt a rabbit that enjoyed children (they took him to visit schools), enjoyed being pet, was okay with being transported, etc.
We are breeders. But we never “sell animals” for material goods. In our University there is an Institution of Pedagogy and Institute of Animal Science. In the Institute of Pedagogy there is a major of “animal-assisted teacher” which is a 2 years old program that gives “specialist” “qualified” teacher degree at the end. The program is only available BSc teachers. All of our rabbits that are suitable for AAI work, are finds owners from the pedagogy students of the University.
The need of “service” animals – regardless to the species – is growing. More and more species has been involved all over the world. BUT we feel the lack of valid scientifically appropriate proven results that deals with the fulfilled stress of the animals. Because we want or not, there are many animals that are forced to participate in AAS (just think about the reptiles that are basically do not enjoy interaction with human at all).
Our aim is to emphasize the need of studies focusing on the welfare of animals involved to any animal assisted activity.
Rabbits’ participation in Animal Assisted Interventions is also an increasing area, however no recommendation is available about how to use them while animal welfare is guaranteed
At our University we started our research on the welfare of rabbits that are participating on AAS in 2017. When we started our work, we used the same methodology as you did. Carefully selected the breeders from whom we bought our first animals. BUT despite of our efforts we recognized that the most of the rabbits just did not feel comfortable in sessions (although the we carefully planned the sessions, with small group of people etc.). We found that the transportation itself is a stressful experience for them (Suba-Bokodi, É.; Nagy, I.; Molnár, M. The Impact of Transportation on the Cortisol Level of Dwarf Rabbits Bred to Ani-mal-Assisted Interventions. Animals 2024, 14, 664.)
Our animals are not commercially available ones. All of our rabbits are selected for tameness during 10 generations and bred for serving in Animal Assisted Interventions. The selection was based on the rabbit’s Human Approach Test according to our study: Suba-Bokodi, É.; Nagy, I.; Molnár, M. Changes in the Stress Tolerances of Dwarf Rabbits in Animal-Assisted Interentions., Appl.Sci. 2022, 12, 6979.
Their ancestors are successfully serving in primary schools, kindergartens and also in elderly taking care homes.
I have to note that the rabbits training for AAS starts just a day after they birth with handling.
You can find it detailed at our article: Changes in the Stress Tolerances of Dwarf Rabbits in Animal-Assisted Interentions., Appl.Sci. 2022, 12, 6979.
In many mammal species, there is a sensitive period during which the nervous system of young animals can be greatly affected. Certain specific stimuli trigger learning processes, which influence the behaviour of the individuals for the rest of their lives, i.e., there is a sensitive period for socialization. Under experimental conditions, such interventions may include handling. The taming effect of handling may have high importance in large-scale production, where working with tamer animals could be advantageous in several ways; for example, they can perform better. Handling is also important for animal therapy because an animal that is less afraid of humans from the beginning will be more suitable for therapy and easier to teach and handle; thus, better results can be achieved [25].
In rabbits, handling consists of touching and holding [26]. Hudson et al. [27] found that even minimal handling carried out in the first week of life reduces shyness. Pongrácz and Altbacker [28] concluded that repeated handling had a positive effect on the behaviour and the welfare of rabbits kept in cages. Kits will become less shy of humans if handling is carried out at times close to suckling. The first week after birth has been shown to be a sensitive period for successful handling. Bilkó et al. [26], Verga et al. [20], and Zucca et al. [29] have found similar results, i.e., handling at an early age has a significant effect on rabbits’ reactivity in behaviour tests. Csatádi [25] examined the effects of early handling under laboratory and natural conditions. He used chinchilla rabbits and New Zealand white rabbits in his research. In the laboratory tests, handling, i.e., touching the kits, was carried out during the first week after birth within half an hour following suckling. At these times, the weight of the kits was typically measured, and their ears were marked: all this took about 3 to 5 min per litter. These “handled” rabbits showed less fear towards humans and became calmer in nature, which continued in their adult life. Another study was conducted to determine if the effects of handling can be specific and whether young rabbits can distinguish between two people. Handling was conducted on two litters (13-13) by two different persons for 7 days. At the age of one month, the litters were divided into two, and the same two persons conducted further handling in an approaching test. The comparison of the animals’ behaviour showed that the person handling the animals did have an effect. The kits showed a preference towards the tester who had been handling them after birth, and both groups were less afraid of people compared to the group which had not been handled at all. Csatádi [25] believes that this can be most probably attributed to olfactory learning. In another experiment, Csatádi [25] examined whether the length of handling influenced its effects. In addition to the usual handling lasting for 3–5 min, 5-s-long handlings, which are actually more applicable in large-scale environments, were conducted with different groups half an hour after suckling and at least 2 h later, and finally, results were compared at the age of one month. They found that only the timing of handling had an effect, whereas its duration did not. Rabbits handled within half an hour after suckling showed less fear towards humans even when they received minimal handling. This may be explained by the fact that tameness comes down to the process of socialisation; thus, the occurrence and the timing of handling are important, but its duration is not [25].
Our aim is to draw attention animal welfare during AAI, to insight to the animals’ side. How animals’ experience these AAI sessions? Is there evidence-based results that deals with the animals fulfilled stress?
We would like to emphasize the need of studies focusing on the welfare of animals involved to any animal assisted activity.
Section 2.1.7 – My program certified a tortoise therapy team here in the U.S. in 2021!
Thank you very much for your comments. I do believe that with your suggestions the quality of the article improved and I highly appreciate for your efforts!